# Development and Validation of a Case-Based Survey Assessing Ethical Decision-Making in Prehospital Resuscitation

**DOI:** 10.3390/healthcare13030267

**Published:** 2025-01-30

**Authors:** Louise Milling, Jeannett Kjær, Oliver B. Sørensen, Sören Möller, Peter M. Hansen, Lars G. Binderup, Caroline Schaffalitzky de Muckadell, Erika F. Christensen, Helle C. Christensen, Annmarie T. Lassen, Dorthe Nielsen, Søren Mikkelsen

**Affiliations:** 1Prehospital Research Unit, Department of Anaesthesiology and Intensive Care, Odense University Hospital, J. B. Winsløws Vej 4, 5000 Odense, Denmark; louisemilling@dadlnet.dk (L.M.); jeannett.kjaer.jorgensen@rsyd.dk (J.K.); peter.martin.hansen@rsyd.dk (P.M.H.); soeren.mikkelsen@rsyd.dk (S.M.); 2Department of Regional Health Research, University of Southern Denmark, Campusvej 55, 5230 Odense M, Denmark; 3Open Patient Data Explorative Network, Odense University Hospital, J. B. Winsløws Vej 21, 5000 Odense C, Denmark; moeller@health.sdu.dk; 4Department of Clinical Research, University of Southern Denmark, Campusvej 55, 5230 Odense M, Denmark; atlassen@health.sdu.dk (A.T.L.); dnielsen@health.sdu.dk (D.N.); 5Philosophy, Department of Design, Media and Educational Science, University of Southern Denmark, Campusvej 55, 5230 Odense M, Denmark; binderup@sdu.dk (L.G.B.); csm@sdu.dk (C.S.d.M.); 6Centre for Prehospital and Emergency Research, Aalborg University Hospital, and Aalborg University, Søndre Skovvej 15, 9000 Aalborg, Denmark; efc@rn.dk; 7Emergency Medical Services, Region North Denmark, Hjulmagervej 20, 9000 Aalborg, Denmark; 8The Prehospital Center, Institute of Clinical Medicine, University of Copenhagen, Ringstedgade 61, 4700 Naestved, Denmark; hcli@regionsjaelland.dk; 9Emergency Medicine Research Unit, Odense University Hospital, J. B. Winsløws Vej 4, 5000 Odense, Denmark; 10Department of Geriatric Medicine, Odense University Hospital, J. B. Winsløws Vej 4, 5000 Odense, Denmark

**Keywords:** cardiac arrest, medical ethics, decision-making, resuscitation, mixed-method

## Abstract

Objectives: Ethical considerations are central to deciding on resuscitation in a prehospital setting. A systematic study of ethical views can enlighten the area and potentially reveal variations in decision-making. We aimed to explore the ethical views on resuscitation and their impact on the reasoning of prehospital healthcare professionals using a qualitative approach and a structured questionnaire. This study describes the validation of a structured questionnaire designed to explore the ethical views on resuscitation and its impact on the reasoning of prehospital healthcare professionals. Methods: This observational cross-sectional study used a mixed-methods approach. The questionnaire included qualitative free-text fields and quantitative scales. Its first version was developed based on data from a systematic review and an ethnographic study. Validation involved face-to-face interviews and a two-round Delphi process with experts in qualitative research, philosophy, epidemiology, and prehospital medicine. The final questionnaire was field-tested among Danish prehospital physicians. Exploratory factor analysis assessed underlying relationships, and Cronbach’s alpha measured internal consistency. Results: 216 out of 380 invited Danish prehospital physicians completed the questionnaire. The ethical aspects addressed in the cases included “do-not-attempt cardiopulmonary resuscitation,” “socioeconomic status,” “quality of life,” “the patient and family’s cultural background,” and “relatives’ emotional reaction.” The questionnaire demonstrated satisfactory internal consistency, with a Cronbach’s alpha of 0.71. Conclusions: The questionnaire was validated as a tool for assessing moral reasoning and variations in perspectives in prehospital decision-making. The survey can be used to assess the moral reasoning and variations therein in prehospital resuscitation decision-making.

## 1. Introduction

Managing out-of-hospital cardiac arrest (OHCA) is vital; early cardiopulmonary resuscitation (CPR) and defibrillation are key to the “chain of survival” [1,2]. Research shows that starting CPR promptly significantly improves the likelihood of return of spontaneous circulation (ROSC) and survival to hospital discharge [2]. Bystander use of automated external defibrillators (AEDs) enhances survival rates, especially with shockable rhythms like ventricular fibrillation [3,4]. When emergency medical services (EMS) arrive, advanced life support (ALS) protocols follow established guidelines [5]. In Denmark, the primary prehospital resource is a two-person ambulance. Additional resources include paramedics in rapid response vehicles or prehospital anaesthesiologists [6]. In Denmark, the decision to stop resuscitation is made by the attending physician, unless the patient has clear signs of death [7].

Decision-making in OHCA is characterised by high complexity. Prehospital clinicians must choose between initiating, continuing, terminating, or withholding treatment, and decisions have to be made on the spot, often with a lack of information and with the presence of external stressors [8]. Choosing to terminate or withhold resuscitation leads to the patient’s imminent death. This complex decision-making has been researched in several qualitative papers describing various factors influencing the decision-making process [9]. These include ethical considerations [10,11] with factors such as the patient’s autonomy, upholding the patient’s dignity, considerations towards the patient’s family, and the physician’s colleagues potentially influencing decisions. Quantitative studies have described variations in the decision-making during prehospital resuscitation due to factors such as the level of experience of the care provider, the characteristics of the patients suffering from OHCA, or the circumstances surrounding the OHCA [12]. It is challenging to investigate medical decision-making using quantitative methods, as surveys can often not encompass the complexity of the decisions. However, it may be feasible when combining conventional surveys with case-based designs. Studies have suggested that case-based designs are useful to assess the prehospital clinicians’ decision-making [13,14], to detect variations in the decision-making and, further, to assess factors that contribute to these variations [15]. Furthermore, combining free-text answers with quantitative variables may allow for a deeper understanding of data and a qualitative exploration of reasoning and ethical considerations in decision-making [16]. Thus, prehospital decision-making about resuscitation is crucial, yet validated tools for evaluating moral reasoning and various perspectives are lacking. Therefore, we aimed at being able to describe ethical reasoning during OHCA treatment and variations in decision-making with a focus on the inclusion of ethical factors by developing and validating a case-based questionnaire.

## 2. Materials and Methods

### 2.1. Setting

Denmark consists of five health regions comprising approximately 5.8 million inhabitants. All regions have a three-tiered prehospital system with a basic resource consisting of two EMTs, a second tier with a paramedic-manned rapid response vehicle, and a third tier with physician- and paramedic-manned Mobile Emergency Care Units (MECU). The latter is dispatched alongside an ambulance or requested as rendezvous by EMTs or paramedics on-scene. Additionally, a Helicopter Emergency Medical Service (HEMS) is available for all regions [6,17]. The questionnaire was aimed at all prehospital physicians employed and operating in the Danish EMS.

### 2.2. Validations Process

A validation study was conducted between November 2021 and February 2023. Figure 1 depicts a flowchart of the validation process. The process consisted of six phases: (1) Developing the cases and the ethical statements and constructing the initial questionnaire, (2) Assessing face validity through structured interviews, (3) Modifying the questionnaire based on the interviews using a modified Delphi process, (4) Testing the modified questionnaire on face validity through interviews, (5) Field testing the final questionnaire among prehospital physicians, (6) Conducting post hoc statistical analyses to assess the questionnaire (see Figure 1). The final questionnaire is available in full in Appendix A. Table 1 depicts the five final sections of the questionnaire and a short description of their content.

#### 2.2.1. Phase 1: Development of Cases and Statements

We conducted a systematic review of the literature concerning non-medical factors, including ethical considerations and challenges in out-of-hospital cardiac arrest [8]. This process has been described elsewhere [8]. These findings were combined with well-known ethical theory from Beauchamp and Childress’ on the four principles of medical ethics [18]. Furthermore, we included qualitative findings from participant observations on prehospital clinicians’ ethical reflections and their encounters with ethical challenges in cardiac arrest in a prehospital setting. Based on these findings, we have developed the statements in Section 2. The first author, LM, conducted participant observations of the prehospital treatment of out-of-hospital cardiac arrest [19]. Findings from these observations were used in the case vignette development.

A 7-point Likert scale was developed for the statements in Section 2 of the questionnaire (see Table 2) (1 = “extremely unimportant”, 7 = “extremely important”) [20]. A 7-point Likert scale was chosen to investigate which ethical considerations the participants deemed important in decision-making. A binary response (terminate/continue resuscitation) was developed for the case vignettes in Section 1 with a mandatory free text field to indicate the reasons for the decision to terminate or continue resuscitation.

#### 2.2.2. Phase 2: Structured Interviews

We included five participants who filled out the questionnaire while a researcher from the research team observed and took field notes during the completion. The participants were asked to think aloud and share their thoughts on the questions, possible answers, and the questionnaire as a whole. Furthermore, the researcher timed the completion and asked follow-up questions.

#### 2.2.3. Phase 3: Evaluating the Statements with Experts

In the survey’s first and second sections, a modified Delphi method was used to choose five cases and 15 statements [21,22]. The modified Delphi method is a group consensus strategy that uses expert opinions and the literature reviews to reach a consensus on a topic. The method is useful when evidence is scarce, as it relies on a collective effort from the expert panel to produce knowledge beyond what they could have produced individually and, in turn, increase content validity [23,24]. The panel consisted of eight Danish experts representing anaesthesiology, prehospital medicine, emergency medicine, philosophy, and qualitative research. The expert panel reviewed the cases based on real-life situations (Section 1). Following two meetings, they agreed on the five cases that they deemed most relevant (see Appendix A). In the second section, the expert panel had to select fifteen statements they considered most relevant in relation to the research question from a list of 30 statements. Statements with less than two votes were discarded. This process was repeated once. The experts then had to rank the ten most relevant statements from least important to most important.

#### 2.2.4. Phase 4: Test by Physicians

After development, initial testing, and subsequent modifications, five physicians tested the questionnaire. The physicians were experienced in cardiology, out-of-hospital cardiac arrest, and ethical challenges but were not part of the final field-testing population. They provided written and verbal feedback on the cases, response options, and the questionnaire as a whole to optimise content and face validity.

#### 2.2.5. Phase 5: Field-Test, Participants, and Setting

All prehospital physicians employed and operating in the Danish EMS were eligible for inclusion. The physicians had to be clinically active in the prehospital field in a MECU, HEMS, or both, and additionally, they needed to work as anaesthesiologists. The questionnaire was distributed electronically. The medical directors of the MECU and HEMS were contacted and asked to distribute a link to the survey via e-mail to all eligible participants. Inclusion was open from April to October 2022

#### 2.2.6. Phase 6: Post Hoc Analyses of Statements Concerning Ethical Considerations (Section 2)

Continuous variables were summarised as median and interquartile range (IQR). An exploratory factor analysis (EFA) was used to identify and assess underlying relationships between variables and determine relevant factors based on patterns in the questionnaire. EFA groups items together when their response patterns are similar across the study population [25]. A Cronbach’s alpha was estimated to assess the internal consistency for each identified factor as well as the questionnaire in its entirety [25]. A low Cronbach’s alpha (0.70) indicates variability in how individuals perceive the significance of items like Age and Dignity, despite their grouping by EFA [26]. Stata (StataCorp. 2021. Stata Statistical Software: Release 17. College Station, TX, USA, StataCorp LLC) was used for statistical analysis [27].

### 2.3. Translation

The final questionnaire (Appendix A) was translated using a modified back-translation process [28]. Using two translators, one of whom was native Danish but fluent in English, and another native in English but fluent in Danish, translated the questionnaire. First, from Danish to English, and subsequently, from English to Danish, to ensure that nuances were not lost. Both translators were acquainted with the study purpose and project specifications [28].

## 3. Results

The results are presented in two parts. The first part describes the development and testing of the questionnaire in its entirety, while the second part presents results from the post hoc statistical analyses focusing on Section 2 of the questionnaire.

### 3.1. Development, Content Validity, and Face Validity

A total of 88 questions were generated. The participants in Phase 2 (structured interviews) found the initial questionnaire too extensive. They were confused regarding the purpose of the survey because of the many ethical dimensions in Section 2 (see Figure 1 for an overview of the phases). Because of this feedback, the questionnaire was reduced using a modified Delphi method. As described previously, the number of items was reduced to a final number of 29 items. The final questionnaire consisted of six overall sections (see Table 1). See Appendix A for the complete questionnaire.

#### 3.1.1. Modified Delphi

In Section 1, the expert panel decided on five cases through discussion during two meetings. These cases covered different areas of ethical decision-making, including do-not-attempt cardiopulmonary resuscitation (DNACPR), socioeconomic status, perceived quality of life, the patient and the family’s cultural background, and relatives’ emotional reactions. One case was considered a neutral case, not including any additional information about the patient, to act as a baseline for decision-making. The ten case vignettes not included in the questionnaire concerned fear of complaints, management of relatives after termination, the influence of guidelines, the dilemma between DNACPR and clinical factors, lack of DNACPR, emotional reactions from relatives, concerns for one’s own safety, and educational purposes.

In Section 2, the expert panel ranked 30 statements and decided on fifteen final statements covering DNACPR, patient autonomy and dignity, in-hospital capacity, level of experience, and relatives’ emotional reactions. See Appendix A for results from the first Delphi round and Appendix A for the final statements in Section 2 of the questionnaire.

#### 3.1.2. Testing of Face Validity of the Final Questionnaire

The questionnaire was tested by five physicians after the reduction in the number of questions. The physicians reported the questionnaire as understandable and believed it covered the most important areas of ethical considerations in prehospital resuscitation. Two participants in the test group suggested adding examples after the statements in Section 2 to ensure that the questions were understood correctly.

### 3.2. Field Test

In the field test, 380 physicians were invited to participate, and 216 responded to the questionnaire (response rate 57%). See Table 2 for an overview of the participants’ characteristics. In Section 1 of the questionnaire, 215 participants wrote one sentence or more in the free-text field concerning the reasoning behind decision-making, and one participant filled in a full stop in all five fields, most likely to circumvent the forced response option.

### 3.3. Item Score-Distribution

The respondents used all response categories. The most used response was “Moderately important,” and the least common “A little unimportant” on the 7-point Likert scale. See Table 3 for the distribution of answers.

### 3.4. Post Hoc Analysis

There were five factors identified in the exploratory factor analysis (see Figure 2) consisting of one to six items. The largest factor contains patient wishes as well as both DNACPR items, physical condition, and quality of life, and showed high consistency with a Cronbach’s alpha of 0.79. The second factor included experience, the expected quality of life, and subjective assessment, but an alpha of just 0.51 was obtained, indicating limited consistency between those three items. The third factor consisted of the risk of complaints and relatives’ emotional reactions, with an acceptable alpha of 0.62. The fourth factor consisted of age and dignity, with very limited consistency and an alpha of 0.11. Finally, the in-hospital occupancy situation was identified as a single-item fifth factor. The whole questionnaire resulted in a satisfactory alpha of 0.71, indicating that the questionnaire can be used as one-dimensional when desired [26].

## 4. Discussion

In this study, we described the development and validation process of a questionnaire designed to investigate the ethical considerations in prehospital resuscitation decision-making. We found that developing a questionnaire covering all dimensions of ethical decision-making would make it too extensive for the respondents. The reduced and final questionnaire had a satisfactory Cronbach’s alpha value, indicating that the questionnaire could be used to describe the same construct, i.e., considerations influencing decision-making.

Using a questionnaire to assess ethical considerations, let alone ethical practice, as was the case in this study, presents certain challenges. Firstly, responses in the questionnaire may not reflect everyday practice [29]. We presumed that the reflections in the mandatory free-text fields under each case vignette could indicate ethical reflections in decision-making. Furthermore, we hypothesised that ethical reflections and reasoning are concomitant with acting in accordance with the conclusions reached via the ethical reflections and reasoning. As such, the questionnaire can be used as a foundation in the research of ethical considerations and reflections but should be supported by research on ethical reflections through field studies. This notion is supported by other studies on ethical sensitivity and attitudes [30]. Secondly, ethical challenges and considerations are often very context dependent and developing a questionnaire encompassing all nuances of ethical considerations in prehospital resuscitation proved difficult. With the commonly found ethical topics included, the questionnaire was too extensive for the participants. This underlines the difficult choices that arise when developing a feasible questionnaire. On the one hand, the researcher aims to cover all areas of a topic but, at the same time, seeks to include enough participants to answer the research questions. We chose to reduce the number of questions to increase the respondents’ willingness to complete the entire questionnaire.

Studies indicate an increase in reliability moving from 2-point scales to 7-point scales [31]. Scales that are too short may not adequately reveal variations in one participant’s response across a large set of questions. In accordance with Taherdoost, we thus settled for a 7-point scale [31]. The choice of a Likert scale comes with a risk of participants avoiding extreme responses, causing a central tendency bias [32]. This was not the case in our study, as participants used the entire scale. It was not possible to statistically test sections of the questionnaire, e.g., the case vignettes. However, they were assessed by test groups that deemed the cases to be realistic and relevant.

We chose a mixed-methods approach and constructed the questionnaire to include both qualitative free-text fields to describe reasoning as well as quantitative scales. The mixed-methods methodology is useful to study complex phenomena such as resuscitation decision-making [33].

This is the first questionnaire aimed at investigating ethics in prehospital resuscitation decision-making. The ethical views in out-of-hospital cardiac arrest have been explored through qualitative studies [8], while measurement on a larger scale, to our knowledge, has not been attempted. This questionnaire facilitates investigation across time, geographical areas, and cultures. Measuring ethical attitudes, reflections, or sensitivity in medicine has been attempted in several studies—however, few with validated instruments. In a narrative review, Kotzee et al. investigated the measurement of medical ethics and highlighted the lack of moral virtue in medical ethics research [34]. As such, most studies focus on the cognitive aspects and investigate the moral reasoning behind actions, not the healthcare professionals’ moral virtues, e.g., empathy, care, or wisdom. This is also the case in our questionnaire, which primarily focuses on reasoning and decision-making. In prehospital resuscitation, these aspects of treatment are of great importance for the patient, and potential variations may lead to disparities in treatment [8,35]. Thus, this questionnaire aims specifically at evaluating potential variations and should not be used to conclude on all nuances of morality in decision-making.

Hebért et al. developed a tool to measure ethical sensitivity in medical students. The authors chose a vignette-based design and found it feasible to detect differences in ethical sensitivity [29]. Even though case vignettes may be useful, a case vignette will not be able to capture all aspects of a real-life case, and it is almost certain that a healthcare professional will not experience a precisely similar situation in reality, which means one cannot generalise from the findings. On the other hand, a vignette that is well-written and realistic can be effective in eliciting a sincere response from participants and be as useful as any other case presentation within the limitations of such approaches [36]. We found it feasible to use case vignettes with mandatory free-text reasoning, as almost all participants provided supplemental explanations to their answers.

This study presents a validated tool for assessing moral reasoning and variations in prehospital decision-making related to OHCA. The tool enables researchers to explore regional and cultural differences in ethical decision-making and supports the development of targeted training programmes for prehospital professionals. Furthermore, it provides a foundation for creating evidence-based guidelines aimed at enhancing ethical decision-making in prehospital care. This study supports future research on ethical decision-making in prehospital settings. The validated questionnaire can be utilised across diverse populations to compare different regions and healthcare systems. However, administering the questionnaire outside Denmark may affect response validity, necessitating further validation. Longitudinal studies can examine how moral reasoning evolves over time or in response to new guidelines and training.

## 5. Conclusions

We developed and validated a mixed-method questionnaire on ethical considerations in prehospital resuscitation decision-making. This questionnaire may be useful in assessing the moral reasoning and variations in prehospital resuscitation decision-making. However, using the questionnaire in other settings than Danish could influence the validity of the responses, and validation processes should be conducted.

## Figures and Tables

**Figure 1 healthcare-13-00267-f001:**
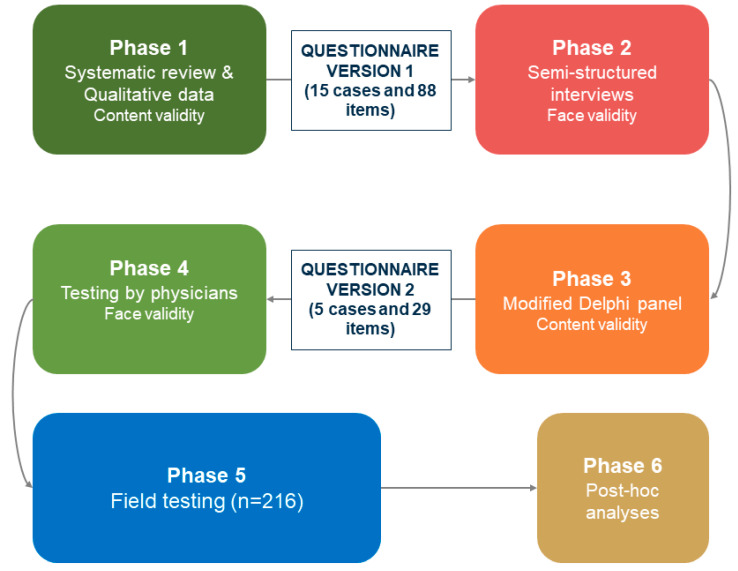
Overview over the phases in the study.

**Figure 2 healthcare-13-00267-f002:**
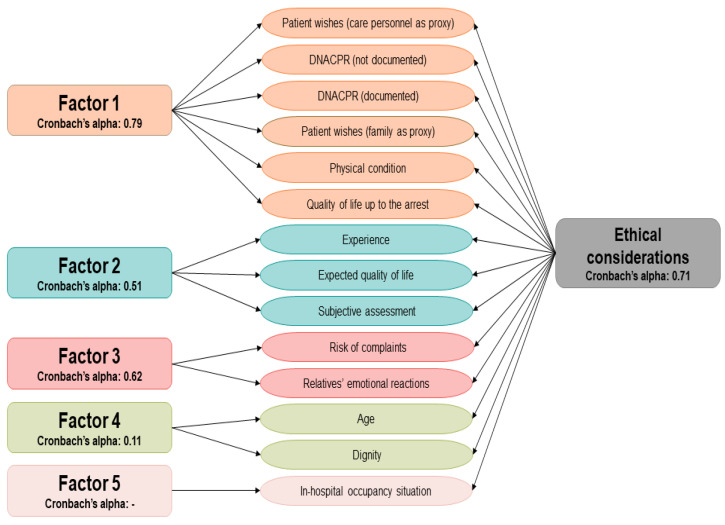
Factors identified in the exploratory factor analysis.

**Table 1 healthcare-13-00267-t001:** An overview of the questionnaire sections.

	Main		Appendix A			Respondent Information
	Appendix A Appendix A	Appendix A Appendix A	Appendix A Appendix A	Appendix A Appendix A	Appendix A Appendix A	Appendix A Appendix A
**Type of questions**	Cases based on real-life OHCA * situations	Statements concerning ethical considerations in prehospital resuscitation	Most reliable sources of information in prehospital resuscitation	The extent of ethical documentation	Need for guidance concerning ethical factors	Demographics
**Response categories**	Binary (Terminate resuscitation/ continue resuscitation) + An obligatory free text field with the reasoning behind the choice to terminate or continue resuscitation	7-point Likert scale from “Extremely unimportant” to “Extremely important” + A free text field allowing mentions of additional considerations	Check-boxes with possible sources of information (e.g., care personnel, relatives, medical records): Choosing three most important	5-point scale from “Every time” to “Never” + a free-field of why/why not	Binary (yes/no)	Age, gender, level of experience, subspecialty

* OHCA: Out-of-hospital cardiac arrest.

**Table 2 healthcare-13-00267-t002:** Demographics of the field test population.

Demographics	Study Population = 216
**Gender, female, *n* (%)**	56 (25.9)
**Age, years, median (IQR *)**	47.5 (33)
**Primary area of work, *n* (%)**	
Intensive Care	49 (22.7)
Clinical anaesthesiology	115 (53.2)
Pain medicine	1 (0.5)
Prehospital	27 (12.5)
Other	1 (0.5)
**Level of experience, *n* (%)**	
<2 years	20 (9.3)
2–5 years	40 (18.5)
6–10 years	50 (23.2)
>10 years	83 (38.4)
**Region, *n* (%) ****	
North Denmark Region	25 (11.6)
Central Denmark Region	75 (34.7)
Region of Southern Denmark	60 (27.8)
Region Zealand	15 (6.9)
Capital Region of Denmark	47 (21.8)

* IQP: Inter-quartile range; ** The total number of responses for region exceeds 216 because some respondents work in more than one region. Consequently, the percentages for this category sum to more than 100%.

**Table 3 healthcare-13-00267-t003:** Distribution of responses (n = 216).

	Extremely Unimportant n (%)	Moderately Unimportant n (%)	A Little Unimportant n (%)	Neither Important nor Unimportant n (%)	A Little Important n (%)	Moderately Important n (%)	Extremely Important n (%)	Missing n (%)
Age	0 (0)	4 (1.9)	4 (1.9)	9 (4.2)	45 (20.8)	107 (49.5)	30 (13.9)	17 (7.9)
Physical condition	0 (0)	0 (0)	3 (1.4)	0 (0)	16 (7.4)	81 (37.5)	99 (45.8)	17 (7.9)
Subjective evaluations	10 (4.6)	19 (8.8)	14 (6.5)	33 (15.3)	72 (33.3)	41 (19)	10 (4.6)	17 (7.9)
QoL before cardiac arrest	2 (0.9)	4 (1.9)	5 (2.3)	14 (6.5)	51 (23.6)	77 (35.7)	45 (20.8)	18 (8.3)
Expected QoL *	2 (0.9)	2 (0.9)	0 (0)	8 (3.7)	19 (8.8)	82 (38)	85 (39.4)	18 (8.3)
DNACPR ** (documented)	0 (0)	0 (0)	2 (0.9)	4 (1.9)	9 (4.2)	37 (17.1)	145 (67.1)	19 (8.8)
DNACPR (verbally, care personnel)	5 (2.3)	11 (5.1)	7 (3.2)	17 (7.9)	57 (26.4)	75 (34.7)	25 (11.6)	19 (8.8)
DNACPR (Not documented)	2 (0.9)	6 (2.8)	7 (3.2)	18 (8.3)	66 (30.6)	77 (35.7)	21 (9.7)	19 (8.8)
Dignity	2 (0.9)	2 (0.9)	2 (0.9)	6 (2.8)	18 (8.3)	57 (26.4)	110 (50.9)	19 (8.8)
DNACPR (verbally, relatives)	5 (2.3)	19 (8.8)	16 (7.4)	27 (12.5)	68 (31.5)	44 (20.4)	18 (8.3)	19 (8.8)
Emotional reactions from relatives	25 (11.6)	44 (20.4)	18 (8.3)	49 (22.7)	50 (23.2)	11 (5.1)	0 (0)	19 (8.8)
Complaints	52 (24.1)	35 (16.2)	20 (9.3)	37 (17.1)	31 (14.4)	19 (8.8)	3 (1.4)	19 (8.8)
Experiences	4 (1.9)	6 (2.8)	4 (1.9)	23 (10.7)	59 (27.3)	58 (26.9)	43 (19.9)	19 (8.8)
In-hospital occupancy	172 (79.6)	13 (6)	2 (0.9)	7 (3.2)	3 (1.4)	0 (0)	0 (0)	19 (8.8)

* Qol: Quality of life; ** DNACPR: Do-not-attempt cardiopulmonary resuscitation.

## Data Availability

Dataset available on request from the authors.

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
