# Peer review of "Development and Validation of a Case-Based Survey Assessing Ethical Decision-Making in Prehospital Resuscitation"

_healthcare, 2025, doi:10.3390/healthcare13030267_

Round 1
Reviewer 1 Report
Comments and Suggestions for Authors
Dear authors,
It is a pleasure to review the manuscript entitled “Development and validation of a case-based survey assessing ethical decision-making in prehospital resuscitation.”
The authors have submitted a manuscript that concerns the ethical consideration, which are important regarding the decisions of continued resuscitation in the prehospital setting. They used a qualitative approach and a structured questionnaire to explore the ethical views on the termination of resuscitation in the field. They used a validation process, which involved face-to-face interviews as well as a two-round Delphi process performed by experts in prehospital medicine, qualitative research, philosophy, and epidemiology. The developed questionnaire was used among Danish prehospital physicians, and the resulting exploratory factor analysis assessed underlying relationships, and Cronbach’s alpha value measured internal consistency. Out of 380 invited physicians, 216 completed the questionnaire, which represents a high rate of completion. Regarding the ethical considerations, the following aspects included not attempting CPR, socioeconomic status, quality of life, cultural background, and emotional reaction of family members. The questionnaire was determined to be internally consistent with a Cronbach’s alpha value of 0.71. The authors conclude that this survey could be used as a tool for determining moral reasoning in the prehospital setting for the determining the discontinuation of resuscitation efforts.
This paper is detailed in its analysis, and the steps taken to develop and analyze the final survey are discussed at great length. For example, in Page 6, Lines 229–232: “one participant filled in a full stop in all five fields, most likely to circumvent the forced response option.”
These are the revisions that I would recommend:
Page 4, Lines 165–168: It may be helpful to include descriptions of the definitions of exploratory factor analysis and Cronbach’s alpha in layman’s terms to clarify the analysis. For example, to my understanding, Age and Dignity in the current study were grouped together into Factor 4 because the exploratory factor analysis grouped them together based on the similarity of their distributions of responses as shown in Table 3. However, the low Cronbach’s alpha of 0.11 for Factor 4 was due to individual respondents' lack of consistency regarding the importance of Age and Dignity. This would indicate that exploratory factor analysis makes groups based on similar aggregate distributions of answers, while Cronbach’s alpha looks at consistency in individual responses. This point could be clarified for the benefit of the non-statistician reader.
Table 1: Consider adding an “e” to “Respons” in the phrase “Respons categories”
Table 2: There is the assumption that total number of responses > 216 for Region because some respondents work in more than one region. As a result, the percentages for Region add up to more than 100%, and this point would be interesting to point out.
Table 3: In the row “Subjective evaluations” and column “Moderately unimportant,” the data point is not aligned in the table, which can make the table somewhat difficult to interpret. Additionally, the title of the table indicates that the table shows a “Spearman’s correlation matrix,” but the table just seems to show the distribution of responses and not Spearman’s correlation values in a matrix format. Additionally, in the abbreviations listed at the bottom of Table 3, the abbreviation of DNACPR is listed as “Do-no-attempt cardiopulmonary resuscitation” instead of “Do-not-attempt cardiopulmonary resuscitation” as it was defined earlier in the paper. Consider revising the table with these changes to better demonstrate the data.
Discussion: The authors’ conclusion is that this questionnaire is “validated as a tool for assessing moral reasoning and variations in perspectives in prehospital decision-making” and “can be used to assess the moral reasoning and variations therein in prehospital resuscitation decision-making.” However, the manuscript does not include an additional analysis regarding the relative importance of the various parameters that have been subjected to the exploratory factor analysis and Cronbach’s alpha analysis. As a result, the reader has no idea regarding the respondent’s ranked preference for various items. For example, it would be interesting to point out that In-hospital occupancy was deemed extremely unimportant to the majority of the respondents. Additionally, it is unknown if Age, Physical Condition, or any other item was important according to the respondents because there is no discussion of which items were important to many of the survey-takers besides the aggregate distributions of responses in Table 3. As a result, “the moral reasoning” cannot be readily seen. I would recommend some more analysis of the data to clarify which items were deemed to be important or unimportant to the prehospital physicians. Additionally, while Section 1 of the survey is described in great detail, there is no reporting of the results of this section or much analysis of the results. Adding a table with a summary of the binary responses or important reasonings behind the respondents’ decisions could help to justify the extensive description of Section 1.
Author Response
Comments 1: Page 4, Lines 165–168: It may be helpful to include descriptions of the definitions of exploratory factor analysis and Cronbach’s alpha in layman’s terms to clarify the analysis. For example, to my understanding, Age and Dignity in the current study were grouped together into Factor 4 because the exploratory factor analysis grouped them together based on the similarity of their distributions of responses as shown in Table 3. However, the low Cronbach’s alpha of 0.11 for Factor 4 was due to individual respondents' lack of consistency regarding the importance of Age and Dignity. This would indicate that exploratory factor analysis makes groups based on similar aggregate distributions of answers, while Cronbach’s alpha looks at consistency in individualresponses. This point could be clarified for the benefit of the non-statistician reader.’
Response 1: Thank you for this thoughtful suggestion. We agree that providing layman’s descriptions of exploratory factor analysis and Cronbach’s alpha will enhance the clarity of the manuscript for a wider audience.
We have included the following explanations in the manuscript: An exploratory factor analysis (EFA) was used to identify and assess underlying relationships between variables and determine relevant factors based on patterns in the questionnaire, EFA groups items together when their response patterns are similar across the study population [25]. A Cronbach’s alpha was estimated to assess the internal consistency for each identified factor as well as the questionnaire in its entirety [25]. A low Cronbach’s alpha (0.70) indicates variability in how individuals perceive the significance of items like Age and Dignity, despite their grouping by EFA[26].
These additions are included on Page 4, Lines 169–176.
Comments 2: Table 1: Consider adding an “e” to “Respons” in the phrase “Respons categories”
Response 2: Thank you for pointing out this typographical error. We have corrected “Respons categories” to “Response categories” in Table 1.
Response categories
The correction has been made in Table 1
Comments 3: Table 2: There is the assumption that total number of responses > 216 for Region because some respondents work in more than one region. As a result, the percentages for Region add up to more than 100%, and this point would be interesting to point out.
Response 3: Thank you for this observation. You are correct that the total number of responses for Region exceeds 216, as some respondents work in multiple regions. To clarify this, we have added a note below Table 2 stating that respondents could select multiple regions, leading to percentages exceeding 100%
**The total number of responses for each Region exceeds 216 because some respondents work in more than one region. Consequently, the percentages for this category sum to more than 100%
The correction has been made in Table 2
Comments 4: Table 3: In the row “Subjective evaluations” and column “Moderately unimportant,” the data point is not aligned in the table, which can make the table somewhat difficult to interpret. Additionally, the title of the table indicates that the table shows a “Spearman’s correlation matrix,” but the table just seems to show the distribution of responses and not Spearman’s correlation values in a matrix format. Additionally, in the abbreviations listed at the bottom of Table 3, the abbreviation of DNACPR is listed as “Do-no-attempt cardiopulmonary resuscitation” instead of “Do-not-attempt cardiopulmonary resuscitation” as it was defined earlier in the paper. Consider revising the table with these changes to better demonstrate the data.
Response 4: Thank you for pointing out these issues with Table 3. We have carefully reviewed the table and made the following revisions to improve its clarity and accuracy:
Data alignment issue:
The misalignment of the data point in the row “Subjective evaluations” and the column “Moderately unimportant” has been corrected to ensure proper alignment, thereby enhancing clarity and interpretability of the table.
The correction has been made in Table 3
Table title:
The title of the table has been updated to accurately reflect the content, as it presents the distribution of responses rather than a Spearman’s correlation matrix
Distribution of responses (n=216)
The correction has been made in Table 3
Abbreviation for DNACPR:
The abbreviation at the bottom of Table 3 has been corrected from “Do-no-attempt cardiopulmonary resuscitation” to “Do-not-attempt cardiopulmonary resuscitation,”
Do-not-attempt cardiopulmonary resuscitation
The correction has been made in Table 3
We appreciate your observations and feedback
Comments 5: Discussion: The authors’ conclusion is that this questionnaire is “validated as a tool for assessing moral reasoning and variations in perspectives in prehospital decision-making” and “can be used to assess the moral reasoning and variations therein in prehospital resuscitation decision-making.” However, the manuscript does not include an additional analysis regarding the relative importance of the various parameters that have been subjected to the exploratory factor analysis and Cronbach’s alpha analysis. As a result, the reader has no idea regarding the respondent’s ranked preference for various items. For example, it would be interesting to point out that In-hospital occupancy was deemed extremely unimportant to the majority of the respondents. Additionally, it is unknown if Age, Physical Condition, or any other item was important according to the respondents because there is no discussion of which items were important to many of the survey-takers besides the aggregate distributions of responses in Table 3. As a result, “the moral reasoning” cannot be readily seen.
I would recommend some more analysis of the data to clarify which items were deemed to be important or unimportant to the prehospital physicians. Additionally, while Section 1 of the survey is described in great detail, there is no reporting of the results of this section or much analysis of the results. Adding a table with a summary of the binary responses or important reasonings behind the respondents’ decisions could help to justify the extensive description of Section 1.
Response 5: Thank you for this important observation. We consider this paper a methodological manuscript describing primarily the construction of the questionnaire. However, we plan on publishing an additional paper discussing the very points you are addressing above. Furthermore, the future paper will include a mixed method analysis of the responses. The responses are indeed very interesting and warrant a longer elaboration than this paper allows for.

Reviewer 2 Report
Comments and Suggestions for Authors
The proposed manuscript “Development and validation of a case-based survey assessing ethical decision-making in prehospital resuscitation” presents the results of a co-creation process and the subsequent validation of a questionnaire to investigate operators’ ethical reasoning in the context of out-of-hospital cardiac arrest, specifically with relevance to the resuscitation maneuvers. There are a couple of parts of the manuscript which could be improved:
1) Introduction chapter is very limited. I suggest adding some explanation about the procedures for OHCA intervention, to better clarify the context also to non-experts in the field, and some additional literature research about state-of-art in optimizing the management of OHCA treatment (e.g. optimization of emergency medical services and automated external defibrillators).
2) The discussion section does not clarify the added value of the proposed study in the field. With reference to the research gap (which should also be stated more clearly), authors should recall the reasoning behind it and assess whether the proposed study was capable of filling it or not, and in the first case the implications for researchers in the field, along with future perspectives, should be described.
Author Response
Comments 1: Introduction chapter is very limited. I suggest adding some explanation about the procedures for OHCA intervention, to better clarify the context also to non-experts in the field, and some additional literature research about state-of-art in optimizing the management of OHCA treatment (e.g. optimization of emergency medical services and automated external defibrillators).
Response 1: Thank you for your valuable feedback. We agree that adding more context on OHCA interventions would significantly improve the Introduction section.
Managing out-of-hospital cardiac arrest (OHCA) is vital, early cardiopulmonary resuscitation (CPR) and defibrillation are key to the "chain of survival" [1,2]. Research shows that starting CPR promptly significantly improves the likelihood of return of spontaneous circulation (ROSC) and survival to hospital discharge [2]. Bystander use of automated external defibrillators (AEDs) enhances survival rates, especially with shockable rhythms like ventricular fibrillation [3,4]. When emergency medical services (EMS) arrive, advanced life support (ALS) protocols follow established guidelines [5]. In Denmark, the primary prehospital resource is a two-person ambulance. Additional resources include paramedics in rapid response vehicles or prehospital anesthesiologists [6]. In Denmark, the decision to stop resuscitation is made by the attending physician, unless the patient has clear signs of death [7].
These additions are included on Page 2, Lines 50–60,
Comments 2: The discussion section does not clarify the added value of the proposed study in the field. With reference to the research gap (which should also be stated more clearly), authors should recall the reasoning behind it and assess whether the proposed study was capable of filling it or not, and in the first case the implications for researchers in the field, along with future perspectives, should be described.
Response 2: Thank you for your insightful feedback. We agree that clearly articulating the research gap, assessing how the study addresses it, and discussing the implications and future perspectives will significantly strengthen the manuscript.
Introduction: Thus, prehospital decision-making about resuscitation is crucial, yet validated tools for evaluating moral reasoning and various perspectives are lacking.
These additions are included on Page 2, Lines 80–82,
Discussion: This study presents a validated tool for assessing moral reasoning and variations in prehospital decision-making related to OHCA. The tool enables researchers to explore regional and cultural differences in ethical decision-making and supports the development of targeted training programmes for prehospital professionals. Furthermore, it provides a foundation for creating evidence-based guidelines aimed at enhancing ethical decision-making in prehospital care. This study supports future research on ethical decision-making in prehospital settings. The validated questionnaire can be utilised across diverse populations to compare different regions and healthcare systems. However, administering the questionnaire outside Denmark may affect response validity, necessitating further validation. Longitudinal studies can examine how moral reasoning evolves over time or in response to new guidelines and training.
These additions are included on Page 9, Lines 347-557,
